# Neural Subgraph Isomorphism Counting

## Abstract

In this paper, we study a new graph learning problem: learning to count subgraph isomorphisms. Although the learning based approach is inexact, we are able to generalize to count large patterns and data graphs in polynomial time compared to the exponential time of the original NP-complete problem. Different from other traditional graph learning problems such as node classification and link prediction, subgraph isomorphism counting requires more global inference to oversee the whole graph. To tackle this problem, we propose a dynamic intermedium attention memory network (DIAMNet) which augments different representation learning architectures and iteratively attends pattern and target data graphs to memorize different subgraph isomorphisms for the global counting. We develop both small graphs ($\leq$ 1,024 subgraph isomorphisms in each) and large graphs ($\leq$ 4,096 subgraph isomorphisms in each) sets to evaluate different models. Experimental results show that learning based subgraph isomorphism counting can help reduce the time complexity with acceptable accuracy. Our DIAMNet can further improve existing representation learning models for this more global problem.

## 1 Introduction

Graphs are general data structures widely used in many applications, including social network analysis, molecular structure analysis, natural language processing and knowledge graph modeling, etc. Learning with graphs has recently drawn much attention as neural network approaches to representation learning have been proven to be effective for complex data structures (Niepert et al., 2016; Kipf & Welling, 2017; Hamilton et al., 2017b; Schlichtkrull et al., 2018; Velickovic et al., 2018; Xu et al., 2019). Most of existing graph representation learning algorithms focus on problems such as node classification, linking prediction, community detection, etc. (Hamilton et al., 2017a). These applications are of more *local* decisions for which a learning algorithm can usually make inferences by inspecting the local structure of a graph. For example, for the node classification problem, after several levels of neighborhood aggregation, the node representation may be able to incorporate sufficient higher-order neighborhood information to discriminate different classes (Xu et al., 2019).

In this paper, we study a more *global* learning problem: learning to count subgraph isomorphisms (counting examples are shown as Figure 1). Although subgraph isomorphism is the key to solve graph representation learning based applications (Xu et al., 2019), tasks of identifying or counting subgraph isomorphisms themselves are also significant and may support broad applications, such as bioinformatics (Milo et al., 2002; Alon et al., 2008), chemoinformatics (Huan et al., 2003), and online social network analysis (Kuramochi & Karypis, 2004). For example, in a social network, we can solve search queries like "groups of people who like X and visited Y-city/state." In a knowledge graph, we can answer questions like "how many languages are there in Africa speaking by people living near the banks of the Nile River?" Many pattern mining algorithms or graph database indexing based approaches have been proposed to tackle subgraph isomorphism problems (Ullmann, 1976; Cordella et al., 2004; He & Singh, 2008; Han et al., 2013; Carletti et al., 2018). However, these approaches cannot be applied to large-scale graphs because of the exponential time complexity.

Thanks to the powerful graph representation learning models which can effectively capture local structural information, we can use a learning algorithm to learn how to count subgraph isomorphisms from a lot of examples. Then the algorithm can scan a large graph and memorize all necessary local information based on a query pattern graph. In this case, although learning based approaches can be inexact, we can roughly estimate the range of the number of subgraph isomorphism. This can already help many applications that do not require exact match or need a more efficient pre-

| Homogeneous | | | Heterogenous | | | | | |
|---|---|---|---|---|---|---|---|---|
| Pattern | Graph | Count | Pattern | Graph | Count | Pattern | Graph | Count |
| △ | ⬚ | 0 | △ | ⬚ | 0 | △ | ⬚ | 0 |
| △ | ◺ | 12 | △ | ◺ | 4 | △ | ◺ | 1 |
| △ | ⧄ | 24 | △ | ⧄ | 6 | △ | ⧄ | 2 |

Figure 1: Subgraph isomorphism counting examples in different settings. The first is the homogeneous graph counting problem. The second is a heterogeneous vertex graph counting problem where there are two types of nodes. The third is a heterogeneous vertex and heterogeneous edge graph counting problem where there are two types of nodes and two types of edges.

processing step. To this end, in addition to trying different representation learning architectures, we develop a dynamic intermedium attention memory network (DIAMNet) to iteratively attend the query pattern and the target data graph to memorize different local subgraph isomorphisms for global counting. To evaluate the learning effectiveness and efficiency, we develop a *small* ($\leq$ 1,024 subgraph isomorphisms in each graph) and a *large* ($\leq$ 4,096 subgraph isomorphisms in each graph) dataset and evaluate different neural network architectures.

Our main contributions are as follows.

• To our best knowledge, this is the first work to model the subgraph isomorphism counting problem as a learning problem, for which both the training and prediction time complexities are polynomial.

• We exploit the representation power of different deep neural network architectures in an end-to-end learning framework. In particular, we provide universal encoding methods for both sequence models and graph models, and upon them we introduce a dynamic intermedium attention memory network to address the more global inference problem for counting.

• We conduct extensive experiments on developed datasets which demonstrate that our framework can achieve good results on both relatively large graphs and large patterns compared to existing studies.

## 2 RELATED WORK

**Subgraph Isomophism Problems**. Given a pattern graph and a data graph, the subgraph isomorphism search aims to find all occurrences of the pattern in the data graph with bijection mapping functions. Subgraph isomorphism is an NP-complete problem among different types of graph matching problems (monomorphism, isomorphism, and subgraph isomorphism). Most subgraph isomorphism algorithms are based on backtracking. They first obtain a series of candidate vertices and update a mapping table, then recursively revoke their own subgraph searching functions to match one vertex or one edge at a time. Ullmann's algorithm (Ullmann, 1976), VF2 (Cordella et al., 2004), and GraphQL (He & Singh, 2008) belong to this type of algorithms. However, it is still hard to perform search when either the pattern or the data graph grows since the search space grows exponentially as well. Some other algorithms are designed based on graph-index, such as gIndex (Yan et al., 2004), which can be used as filters to prune out many unnecessary graphs. However, graph-index based algorithms have a problem that the time and space in indexing also increase exponentially with the growth of the graphs (Sun et al., 2012). TurboISO (Han et al., 2013) and VF3 (Carletti et al., 2018) add some weak rules to find candidate subregions and then call the recursive match procedure on subregions. These weak rules can significantly reduce the searching space in most cases.

**Graph Representation Learning**. Graph (or network) representation learning can be directly learning an embedding vector of each graph node (Perozzi et al., 2014; Tang et al., 2015; Grover & Leskovec, 2016). This approach is not easy to generalize to unseen nodes. On the other hand, graph neural networks (GNNs) (Battaglia et al., 2018) provide a solution to representation learning for nodes which can be generalized to new graphs and unseen nodes. Many graph neural networks have been proposed since 2005 (Gori et al., 2005; Scarselli et al., 2005) but rapidly developed in recent years. Most of them focus on generalizing the idea of convolutional neural networks for general graph data structures (Niepert et al., 2016; Kipf & Welling, 2017; Hamilton et al., 2017b; Velickovic et al., 2018) or relational graph structures with multiple types of relations (Schlichtkrull et al., 2018).

More recently, Xu et al. (2019) propose a graph isomorphism network (GIN) and show its discriminative power. Others use the idea of recurrent neural networks (RNNs) which are originally proposed to deal with sequence data to work with graph data (Li et al., 2016; You et al., 2018). Interestingly, with external memory, sequence models can work well on complicated tasks such as language modeling (Sukhbaatar et al., 2015; Kumar et al., 2016) and shortest path finding on graphs (Graves et al., 2016). There is another branch of research called graph kernels (Vishwanathan et al., 2010; Shervashidze et al., 2011; Yanardag & Vishwanathan, 2015; Togninalli et al., 2019; Chen et al., 2019) which also convert graph isomorphism to a similarity learning problem. However, they usually work on small graphs and do not focus on subgraph isomorphism identification or counting problems.

## 3 PRELIMINARIES

We begin by introducing the subgraph isomophism problems and then provide the general idea of our work by analyzing the time complexities of the problems.

### 3.1 PROBLEM DEFINITION

Traditionally, the subgraph isomorphism problem is defined between two simple graphs or two directed simple graphs, which is an NP-complete problem. We generalize the problem to a counting problem over directed heterogeneous multigraphs, whose decision problem is still NP-complete.

A graph or a pattern is defined as $\mathcal{G} = (\mathcal{V}, \mathcal{E}, \mathcal{X}, \mathcal{Y})$ where $\mathcal{V}$ is the set of vertices. $\mathcal{E} \subseteq \mathcal{V} \times \mathcal{V}$ is the set of edges, $\mathcal{X}$ is a label function that maps a vertex to a vertex label, and $\mathcal{Y}$ is a label function that maps an edge to a set of edge labels. We use an edge with a set of edge labels to represent multiedges with the same source and the same target for clarity. That is, there are no two edges in a graph such that they have the same source, same target, and the same edge label. To simplify the statement, we assume $\mathcal{Y}((u, v)) = \phi$ if $(u, v) \notin \mathcal{E}$.

In this paper, we discuss isomorphic mappings that preserve graph topology, vertex labels and edge labels, but not vertex ids. More precisely, a pattern $\mathcal{G}_P = (\mathcal{V}_P, \mathcal{E}_P, \mathcal{X}_P, \mathcal{Y}_P)$ is *isomorphic* to a graph $\mathcal{G}_G = (\mathcal{V}_G, \mathcal{E}_G, \mathcal{X}_G, \mathcal{Y}_G)$ if there is a bijection $f : \mathcal{V}_G \to \mathcal{V}_P$ such that:

- $\forall v \in \mathcal{V}_G, \mathcal{X}_G(v) = \mathcal{X}_P(f(v))$,
- $\forall v \in \mathcal{V}_P, \mathcal{X}_P(v) = \mathcal{X}_G(f^{-1}(v))$,
- $\forall (u, v) \in \mathcal{E}_G, \mathcal{Y}_G((u, v)) = \mathcal{Y}_P((f(u), f(v)))$,
- $\forall (u, v) \in \mathcal{E}_P, \mathcal{Y}_P((u, v)) = \mathcal{Y}_G((f^{-1}(u), f^{-1}(v)))$.

Furthermore, $\mathcal{G}_P$ being isomorphic to a graph $\mathcal{G}_G$ is denoted as $\mathcal{G}_P \simeq \mathcal{G}_G$ and the function $f$ is named as an *isomorphism*.

A pattern $\mathcal{G}_P = (\mathcal{V}_P, \mathcal{E}_P, \mathcal{X}_P, \mathcal{Y}_P)$ is *isomorphic to a subgraph* $\mathcal{G}'_G = (\mathcal{V}'_G, \mathcal{E}'_G, \mathcal{X}_G, \mathcal{Y}_G)$ of a graph $\mathcal{G}_G = (\mathcal{V}_G, \mathcal{E}_G, \mathcal{X}_G, \mathcal{Y}_G)$: $\mathcal{V}'_G \subseteq \mathcal{V}_G, \mathcal{E}'_G \subseteq \mathcal{E}_G \cap (\mathcal{V}'_G \times \mathcal{V}'_G)$ if $\mathcal{G}_P \simeq \mathcal{G}'_G$. The bijection function $f : \mathcal{V}'_G \to \mathcal{V}_P$ is named as a *subgraph isomorphism*.

The subgraph isomorphism counting problem is defined as to find the number of all different subgraph isomorphisms between a pattern graph $\mathcal{G}_P$ and a graph $\mathcal{G}_G$. Examples are shown in Figure 1.

### 3.2 GENERAL IDEA

Intuitively, we need to compute $\mathcal{O}(Perm(|\mathcal{V}_G|, |\mathcal{V}_P|) \cdot d^{|\mathcal{V}_P|})$ to solve the subgraph isomorphism counting problem by enumeration, where $Perm(n, k) = \frac{n!}{(n-k)!}$, $|\mathcal{V}_G|$ is the number of graph nodes, $|\mathcal{V}_P|$ is the number of pattern nodes, $d$ is the maximum degree. The first subgraph isomorphism algorithm, Ullmann's algorithm (Ullmann, 1976), reduces the seaching time to $\mathcal{O}(|\mathcal{V}_P|^{|\mathcal{V}_G|} \cdot |\mathcal{V}_G|^2)$. If the pattern and the graph are both small, the time is acceptable because both of two factors are not horrendously large. However, since the computational cost grows exponentially, it is impossible to count as either the graph size or the pattern size increases. If we use neural networks to learn distributed representations for $\mathcal{V}_G$ and $\mathcal{V}_P$ or $\mathcal{E}_G$ and $\mathcal{E}_P$, we can reduce the complexity to $\mathcal{O}(|\mathcal{V}_P| \cdot |\mathcal{V}_G| + |\mathcal{V}_G|^2)$ or $\mathcal{O}(|\mathcal{E}_P| \cdot |\mathcal{E}_G| + |\mathcal{E}_G|^2)$ via source attention and self-attention. Assuming that we can further learn a much higher level abstraction without loss of representation power for $\mathcal{G}_P$, then

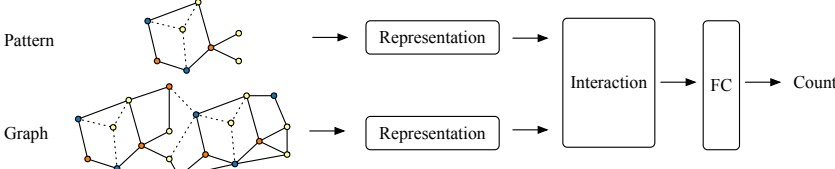

Figure 2: General framework of neural subgraph isomorphism counting models.

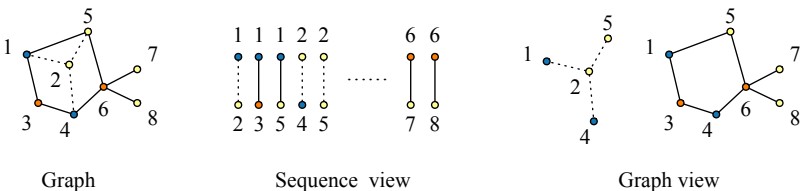

Figure 3: Two views of different encoding methods. There are three labels of vertices (marked as orange, blue, and yellow) and two labels of edges (marked as solid and dashed lines). The numbers affixed show one possible set of ids for the vertices.

the computational cost can be further reduced to $\mathcal{O}(|\mathcal{V}_G|^2)$ or $\mathcal{O}(|\mathcal{E}_G|^2)$. However, the complexity of the latter framework is still not acceptable when querying over large graphs. If we do not consider self-attention, the computational cost will be $\mathcal{O}(|\mathcal{V}_G|)$ or $\mathcal{O}(|\mathcal{E}_G|)$, but missing the self-attention will hurt the performance. In this work, we hope to use attention mechanism and additional memory networks to further reduce the complexity compared with $\mathcal{O}(|\mathcal{V}_G|^2)$ or $\mathcal{O}(|\mathcal{E}_G|^2)$ while keeping the performance acceptable on the counting problem.

## 4 METHODOLOGIES

A graph (or a pattern) can be represented as a sequence of edges or a series of adjacent matrices and vertex features. For sequence inputs we can use CNNs (Kim, 2014), RNNs such as GRU (Cho et al., 2014), or Transformer-XL (Dai et al., 2019) to extract high-level features. While if the inputs are modeled as series of adjacent matrices and vertex features, we can use RGCN (Schlichtkrull et al., 2018) to learn vertex representations with message passing from neighborhoods. After obtaining the pattern representation and the graph representation, we feed them into an interaction module to extract the correlated features from each side. Then we feed the output context of the interaction module into a fully-connected layer to make predictions. A general framework is shown in Figure 2 and the difference between sequence encoding and graph encoding is shown in Figure 3.

### 4.1 SEQUENCE MODELS

#### 4.1.1 SEQUENCE ENCODING

In sequence models, the minimal element of a graph (or a pattern) is an edge. By definition, at least three attributes are required to identify an edge $e$, which are the source vertex id $u$, the target vertex id $v$, and its edge label $y \in \mathcal{Y}(e)$. We further add two attributes of vertices' labels to form a 5-tuple $(u, v, \mathcal{X}(u), y, \mathcal{X}(v))$ to represent an edge $e$, where $\mathcal{X}(u)$ is the source vertex label and $\mathcal{X}(v)$ is the target vertex label. A list of 5-tuple is referred as a code. We follow the order defined in gSpan (Yan & Han, 2002) to compare pairs of code lexicographically; the detailed definition is given in Appendix A. The minimum code is the code with the minimum lexicographic order with the same elements. Finally, each graph can be represented by the corresponding minimum code, and vice versa.

Given that a graph is represented as a minimum code, or a list of 5-tuples, the next encoding step is to encode each 5-tuple into a vector. Assuming that we know the max values of $|\mathcal{V}|, |\mathcal{X}|, |\mathcal{Y}|$

in a dataset in advance, we can encode each vertex id $v$, vertex label $x$, and edge label $y$ into $B$-nary digits, where $B$ is the base and each digit $d \in \{0, 1, \cdots, B - 1\}$. It is easy to replace each digit with a one-hot vector so that each 5-tuple can be vectorized as a multi-hot vector which is the concatenation of one-hot vectors. The length of the multi-hot vector of a 5-tuple is $B \times (2 \cdot \lceil \log_B(Max(|\mathcal{V}|)) \rceil + 2 \cdot \lceil \log_B(Max(|\mathcal{X}|)) \rceil + \lceil \log_B(Max(|\mathcal{Y}|)) \rceil)$. Then minimum is achieved when $B = 2$ and $d_e = 2 \times (2 \cdot \lceil \log_2(Max(|\mathcal{V}|)) \rceil + 2 \cdot \lceil \log_2(Max(|\mathcal{X}|)) \rceil + \lceil \log_2(Max(|\mathcal{Y}|)) \rceil)$. Then we can easily calculate the graph dimension $d_g$ and the pattern dimension $d_p$. Furthermore, the minimum code can be encoded into a multi-hot matrix, $\boldsymbol{G} \in \mathbb{R}^{|\mathcal{E}_G| \times d_g}$ for a graph $\mathcal{G}_G$ or $\boldsymbol{P} \in \mathbb{R}^{|\mathcal{E}_P| \times d_p}$ for a pattern $\mathcal{G}_P$ according to this encoding method.

This encoding method can be extended when we have larger values of $|\mathcal{V}|, |\mathcal{X}|, |\mathcal{Y}|$. A larger value, e.g., $|\mathcal{V}|$, only increases the length of one-hot vectors corresponding to its field. Therefore, we can regard new digits as the same number of zeros in previous data. As long as we process previous one-hot vectors carefully to keep these new dimensions from modifying the original distributed representations, we can also extend these multi-hot vectors without affecting previous models. A simple but effective way is to initialize additional new weights related to new dimensions as zeros.

### 4.1.2 SEQUENCE NEURAL NETWORKS

Given the encoding method in Section 4.1.1, we can simply embed graphs as multi-hot matrices. Then we can use general strategies of sequence modeling to learn dependencies among edges in graphs.

**Convolutional Neural Networks (CNNs)** have been proved to be effective in sequence modeling (Kim, 2014). In our experiments, we apply multiple layers of the convolution operation to obtain a sequence of high-level features.

**Recurrent Neural Networks (RNNs)**, such as GRU (Cho et al., 2014), are widely used in many sequence modeling tasks.

**Transformer-XL (TXL)** (Dai et al., 2019) is a variant of the Transformer architecture (Vaswani et al., 2017) and enables learning long dependencies beyond a fixed length without disrupting temporal coherence. Unlike the original autoregressive settings, in our model the Transformer-XL encoder works as a feature extractor, in which the attention mechanism has a full, unmasked scope over the whole sequence. However, its computational cost grows quadratically with the size of inputs, so the tradeoff between performance and efficiency would be considered.

## 4.2 GRAPH MODELS

### 4.2.1 GRAPH ENCODING

In graph models, each vertex has a feature vector and edges are used to pass information from its source to its sink. GNNs do not need vertex ids and edge ids explicitly because the adjacency information is included in a adjacent matrix. As explained in Section 4.1.1, we can vectorize vertex labels into multi-hot vectors as vertex features. In a simple graph or a simple directed graph, the adjacent information can be stored in a sparse matrix to reduce the memory usage and improve the computation speed. As for heterogeneous graphs, behaviors of edges should depend on edge labels. RGCNs have relation-specific transformations so that each edge label and topological information are mixed into the message to the sink. We follow this method and use basis-decomposition for parameter sharing (Schlichtkrull et al., 2018).

### 4.2.2 GRAPH NEURAL NETWORKS

**Relational Graph Convolutional Networks (RGCNs)** (Schlichtkrull et al., 2018) are developed specifically to handle multi-relational data in realistic knowledge bases. Each relation corresponds to a transformation matrix to transform relation-specific information from a neighbor to the center vertex. Two decomposition methods are proposed to address the rapid growth in the number of parameters with the number of relations: basis-decomposition and block-diagonal-decomposition. We use the first method, which is equivalent to the MLPs in GIN (Xu et al., 2019). The original RGCN uses the mean aggregator, but Xu et al. (2019) find that the sum-based GNNs can capture graph structures better. We implement both and named them as RGCN and RGCN-SUM respectively.

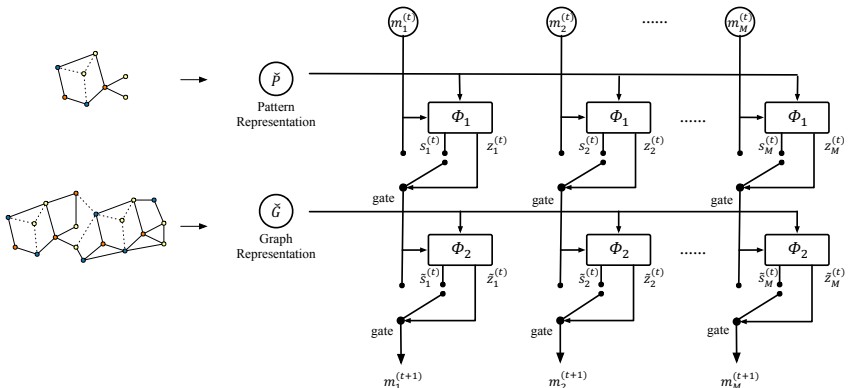

Figure 4: Illustration of dynamic intermedium attention memory network (DIAMNet). $\Phi_1$ represents Eqs. (1) and (2), $\Phi_2$ represents Eqs. (4) and (5), and two types of gates are Eqs. (3) and (6).

### 4.3 DYNAMIC INTERMEDIUM ATTENTION MEMORY NETWORK

After obtaining a graph representation $\check{G}$ and a pattern representation $\check{P}$ from a sequence model or a graph model where their column vectors are $d$-dimensional, we feed them as inputs of interaction layers to extract the correlated context between the pattern and the graph. A naive idea is to use attention modules (Bahdanau et al., 2015) to model interactions between these two representations and interactions over the graph itself. However, this method is not practical due to its complexity, up to $\mathcal{O}(|\mathcal{E}_P| \cdot |\mathcal{E}_G| + |\mathcal{E}_G|^2)$ for sequence modeling and $\mathcal{O}(|\mathcal{V}_P| \cdot |\mathcal{V}_G| + |\mathcal{V}_G|^2)$ for graph modeling.

To address the problem of high computational cost in the attention mechanism, we propose the Dynamic Intermedium Attention Memory Network (DIAMNet), using an external memory as an intermedium to attend both the pattern and the graph in order. To make sure that the memory has the knowledge of the pattern while attending the graph and vice-versa, this dynamic memory is designed as a gated recurrent network as shown in Figure 4. Assuming that the memory size is $M$ and we have $T$ recurrent steps, the time complexity is decreased into $\mathcal{O}(T \cdot M \cdot (|\mathcal{E}_P| + |\mathcal{E}_G|))$ or $\mathcal{O}(T \cdot M \cdot (|\mathcal{V}_P| + |\mathcal{V}_G|))$, which means the method can be easily applied to large-scale graphs.

The external memory is divided into $M$ blocks $\{\boldsymbol{m}_1, ..., \boldsymbol{m}_M\}$, where $\boldsymbol{m}_j \in \mathbb{R}^d$. At each time step $t$, $\{\boldsymbol{m}_j\}$ is updated by the pattern and the graph in order via multi-head attention mechanism (Vaswani et al., 2017). Specifically, the update equations of our DIAMNet are given by:

$$\boldsymbol{s}_j^{(t)} = MultiHead(\boldsymbol{m}_j^{(t)}, \check{\boldsymbol{P}}, \check{\boldsymbol{P}}) \tag{1}$$

$$\boldsymbol{z}_j^{(t)} = \sigma(\boldsymbol{U}_P \boldsymbol{m}_j^{(t)} + \boldsymbol{V}_P \boldsymbol{s}_j^{(t)}) \tag{2}$$

$$\overline{\boldsymbol{s}}_j^{(t)} = \boldsymbol{z}_j^{(t)} \odot \boldsymbol{m}_j^{(t)} + (1 - \boldsymbol{z}_j^{(t)}) \odot \boldsymbol{s}_j^{(t)} \tag{3}$$

$$\widetilde{\boldsymbol{s}}_j^{(t)} = MultiHead(\overline{\boldsymbol{s}}_j^{(t)}, \check{\boldsymbol{G}}, \check{\boldsymbol{G}}) \tag{4}$$

$$\widetilde{\boldsymbol{z}}_j^{(t)} = \sigma(\boldsymbol{U}_G \overline{\boldsymbol{s}}_j^{(t)} + \boldsymbol{V}_G \widetilde{\boldsymbol{s}}_j^{(t)}) \tag{5}$$

$$\boldsymbol{m}_j^{(t+1)} = \widetilde{\boldsymbol{z}}_j^{(t)} \odot \overline{\boldsymbol{s}}_j^{(t)} + (1 - \widetilde{\boldsymbol{z}}_j^{(t)}) \odot \widetilde{\boldsymbol{s}}_j^{(t)} \tag{6}$$

Here $MultiHead$ is the attention method described in (Vaswani et al., 2017), $\sigma$ represents the logistic sigmoid function, $\overline{\boldsymbol{s}}_j$ is the intermediate state of the $j^{th}$ block of memory that summarizes information from the pattern, and $\widetilde{\boldsymbol{s}}$ for information from both the pattern and the graph. $\boldsymbol{z}_j$ and $\widetilde{\boldsymbol{z}}_j$ are two gates designed to control the updates on the states in the $j^{th}$ block. $\boldsymbol{U}_P, \boldsymbol{V}_P, \boldsymbol{U}_G, \boldsymbol{V}_G \in \mathbb{R}^{d \times d}$ are trainable parameters.

## 5 EXPERIMENTS

In this section, we report our major experimental results. More results can be found in the Appendix.

## 5.1 DATASETS

In order to train and evaluate our neural models for the subgraph isomorphism counting problem, we need to generate enough graph-pattern data. As there's no special constraint on the pattern, the pattern generator may produce any connected multigraph without identical edges, i.e., parallel edges with identical label. In contrast, the ground truth number of subgraph isomorphisms must be tractable in our synthetic graph data. Therefore, our graph generator first generates multiple disconnected components, possibly with some subgraph isomorphisms. We use the idea of neighborhood equivalence class (NEC) in TurboISO (Han et al., 2013) to control the necessary conditions of a subgraph isomorphism in the graph generation process. The detailed algorithms are shown in Appendix B. Then the generator merges these components into a larger graph and ensures that there is no more subgraph isomorphism generated in the merge process. The subgraph isomorphism search can be done during these components subgraphs in parallel.

Using the pattern generator and the graph generator above, we can generate many patterns and graphs for neural models. We are interested in follow research questions: whether sequence models and graph convolutional networks can perform well given limited data, whether their running time is acceptable, and whether memory can help models make better predictions even faced with a NP-complete problem. To evaluate different neural architectures and different prediction networks, we generate two datasets in different graph scales and the statistics are reported in Table 1. There are 187 unique patterns in whole pairs, where 75 patterns belong to the *small* dataset, 122 patterns belong to the *large* dataset. Target data graphs are not required similar so they are generated randomly. The generation details are reported in Appendix C.

Table 1: Statistics of the datasets.

|  | #Training | #Dev | #Test | Mean($|\mathcal{V}_G|$) | Mean($|\mathcal{E}_G|$) | Counts |
|---|---|---|---|---|---|---|
| Small | 398,088 | 49,761 | 49,761 | 32.6 | 76.3 | $\{c \in \mathbb{N} | c \leq 1,024\}$ |
| Large | 316,224 | 39,528 | 39,528 | 240.0 | 560.0 | $\{c \in \mathbb{N} | c \leq 4,096\}$ |

## 5.2 IMPLEMENTATION DETAILS

Instead of directly feeding multi-hot encoding vectors into representation modules, we use two simple linear layers separately to transform graph multi-hot vectors and pattern multi-hot vectors to lower-dimensional, distributed ones. To improve the efficiency, we also add a filtering layer to filter out irrelevant parts before all representation modules. The details of this filter layer is shown in Section D.1.

### 5.2.1 REPRESENTATION MODELS

In our experiments, we implemented five different representation models: (1) **CNN** is a 3-layer convolutional layers followed by max-pooling layers. The convolutional kernels are 2,3,4 respectively and strides are 1. The pooling kernels are 2,3,4 and strides are 1. (2) **RNN** is a simple 3-layer GRU model. (3) **TXL** is a 6-layer Transformer encoder with additional memory. (4) **RGCN** is a 3-layer RGCN with the basis decomposition. We follow the same setting in that ordinal paper to use mean-pooling in the message propagation part. (5) **RGCN-SUM** is a modification of RGCN to replace the mean-pooling with sum-pooling.

### 5.2.2 INTERACTION NETWORKS

After getting a graph representation $\check{G}$ and a pattern representation $\check{P}$ from the representation learning modules, we feed them into the following different types of interaction layers for comparison.

**SumPool**: A simple sum-pooling is applied for $\check{G}$ and $\check{P}$ to obtain $\check{g}$ and $\check{p}$, and the model sends $Concate(\check{g}, \check{p}, \check{g} - \check{p}, \check{g} \odot \check{p})$ with the graph size and the pattern size information into the next fully connected (FC) layers.

**MeanPool**: Similar settings as SumPool, but to replace the pooling method with mean-pooling.

**MaxPool**: Similar settings as SumPool, but to replace the pooling method with max-pooling.

**AttnPool**: We want to use attention modules without much computational cost so the self-attention is not acceptable. We simplify the attention by first applying a pooling mechanism for the pattern graph and then use the pooled vector to perform attention over the data graph rather than simply perform pooling over it. Other settings are similar with pooling methods. The detailed information is provided in Appendix D.2. We only report results of mean-pooling based attention, because it is the best of the three variants.

**DIAMNet**: We compare the performance and efficiency of our DIAMNet proposed in Section 4.3 with above interaction networks. The initialization strategy we used is shown in Appendix D.3. And we feed the whole memory with size information into the next FC layers.

## 5.3 EXPERIMENT SETTINGS

For fair comparison, we set embedding dimensions, dimensions of all representation models, and the numbers of filters all 64. The segment size and memory size in TXL are also 64 due to the computation complexity. The length of memory is fixed to 4 and the number of recurrent steps is fixed to 3 in the DIAMNet for both *small* and *large* datasets. We use the mean squared error (MSE) to train models and evaluate the validation set to choose best models. The optimizer is Adam with learning rate 0.001. L2 penalty is added and the coefficient is set as 0.001. To avoid gradient explosion and overfitting, we add gradient clipping and dropout with a dropout rate 0.2. We use $Leaky\_ReLU$ as activation functions in all modules. Due to the limited number of patterns, the representation module for patterns are easy to overfit. Therefore, we use the same module with shared parameters to produce representation for both the pattern and the graph. We also find that using curriculum learning (Bengio et al., 2009) can help models to converge better. Hence, all models in Table 3 are fine-tuned based on the best models in *small* in the same settings. Training and evaluating were finished on one single NVIDIA GTX 1080 Ti GPU under the PyTorch framework.

## 5.4 EVALUATION METRICS

As we model this subgraph isomorphism counting problem as a regression problem, we use common metrics in regression tasks, including the root mean square error (RMSE) and the mean absolute error (MAE). In this task, negative predictions are meaningless, so we only evaluate $ReLU(\widehat{Y_i})$ as final prediction results. Considering that about 75% of countings are 0's in our dataset, we also use evaluation metrics for the binary classification to analyze behaviors of different models. We report F1 scores for both zero data (F1$_{\text{zero}}$) and nonzero data (F1$_{\text{nonzero}}$). Two trivial baselines, **Zero** that always predicts 0 and **Avg** that always predicts the average counting of training data, are also used in comparison.

## 5.5 RESULTS AND ANALYSIS

We first report results for *small* dataset in Table 2 and results for *large* dataset in Table 3. In addition to the trivial all-zero and average baselines and other neural network learning based baselines, we are also curious about to what extent our neural models can be faster than traditional searching algorithms. Therefore, we also compare the running time. Considering the graph generation strategy we used, we decide to compare with VF2 algorithm (Cordella et al., 2004) to avoid unnecessary interference from the similar searching strategy. From the experiments, we can draw following observations and conclusions.

**Comparison of different representation architectures.** As shown in Table 2, in general, graph models outperform most of the sequence models but cost more time to do inference. CNN is the worst model for the graph isomorphism counting problem. The most possible reason is that the sequence encoding method is not suitable for CNN. The code order does not consider the connectivity of adjacent vertices and relevant label information. Hence, convolutional operations and pooling operations cannot extract useful local information but may introduce much noise. From results of the *large* dataset, we can see that F1$_{\text{nonzero}}$=0.180 is even worse than others. In fact, we find that CNN always predicts 0 for large graphs. RNN and TXL are widely used in modeling sequences in many applications. The two models with simple pooling can perform well. We note that RNN with sum-pooling is better than TXL with memory. RNN itself holds a memory but TXL also has much longer memory. However, the memory in RNN can somehow memorize all information that

Table 2: Results of different models on the *small* dataset. Time is evaluated on the whole test set.

| Models | | RMSE | MAE | $F1_{zero}$ | $F1_{nonzero}$ | Time (sec) |
|---|---|---|---|---|---|---|
| | | | | Test | | |
| CNN | SumPool | 55.429 | 11.057 | 0.807 | 0.397 | 0.29 |
| | MeanPool | 57.298 | 10.517 | 0.821 | 0.475 | 0.27 |
| | MaxPool | 47.353 | 11.212 | 0.832 | 0.539 | 0.27 |
| | AttnPool | 55.963 | 12.717 | 0.796 | 0.340 | 0.54 |
| | DIAMNet | **34.448** | **6.953** | **0.884** | **0.753** | 0.90 |
| RNN | SumPool | 29.955 | 5.740 | 0.908 | 0.819 | 0.55 |
| | MeanPool | 31.010 | 6.447 | 0.912 | 0.833 | 0.54 |
| | MaxPool | 30.824 | 6.236 | 0.869 | 0.690 | 0.58 |
| | AttnPool | 31.857 | 6.025 | 0.899 | 0.793 | 0.79 |
| | DIAMNet | **29.743** | **5.547** | **0.927** | **0.877** | 1.15 |
| TXL | SumPool | 34.391 | 7.042 | 0.899 | 0.798 | 1.98 |
| | MeanPool | 32.569 | **6.656** | 0.882 | 0.754 | 1.98 |
| | MaxPool | 65.152 | 30.289 | 0.572 | 0.669 | 1.99 |
| | AttnPool | 37.721 | 7.426 | 0.869 | 0.719 | 2.22 |
| | DIAMNet | **31.649** | 6.680 | **0.900** | **0.811** | 2.48 |
| RGCN | SumPool | 32.414 | 6.578 | 0.900 | 0.796 | 6.05 |
| | MeanPool | 33.829 | 7.152 | 0.878 | 0.735 | 5.99 |
| | MaxPool | 50.851 | 9.707 | 0.869 | 0.704 | 6.04 |
| | AttnPool | 32.526 | 6.523 | 0.870 | 0.697 | 6.27 |
| | DIAMNet | **28.712** | **5.782** | **0.918** | **0.868** | 6.69 |
| RGCN-SUM | SumPool | 22.379 | 3.958 | 0.920 | 0.845 | 5.78 |
| | MeanPool | 22.483 | 4.254 | 0.903 | 0.803 | 5.81 |
| | MaxPool | 42.434 | 7.900 | 0.874 | 0.709 | 6.08 |
| | AttnPool | 24.875 | 5.131 | 0.840 | 0.569 | 6.49 |
| | DIAMNet | **21.734** | **3.853** | **0.924** | **0.864** | 6.55 |
| Zero | | 67.195 | 13.716 | 0.761 | 0.0 | - |
| Avg | | 65.780 | 21.986 | 0.0 | 0.557 | - |
| VF2 | | 0.0 | 0.0 | 1.0 | 1.0 | ∼120 |

Table 3: Results of different models on the *large* dataset. Time is evaluated on the whole test set.

| Model | | RMSE | MAE | $F1_{zero}$ | $F1_{nonzero}$ | Time (sec) |
|---|---|---|---|---|---|---|
| | | | | Test | | |
| CNN | MaxPool | 234.215 | 33.859 | 0.787 | 0.180 | 2.01 |
| | DIAMNet | 193.410 | 29.726 | 0.831 | 0.523 | 5.82 |
| RNN | SumPool | 136.031 | 20.234 | 0.905 | 0.799 | 16.40 |
| | DIAMNet | 134.786 | 21.595 | 0.911 | 0.844 | 19.98 |
| TXL | MeanPool | 182.637 | 35.749 | 0.842 | 0.587 | 61.83 |
| | DIAMNet | 167.480 | 26.954 | 0.881 | 0.749 | 67.95 |
| RGCN | SumPool | 149.060 | 23.051 | 0.927 | 0.864 | 25.74 |
| | DIAMNet | 147.511 | 24.398 | 0.926 | **0.883** | 29.63 |
| RGCN-SUM | SumPool | 125.022 | 16.551 | 0.903 | 0.788 | 25.14 |
| | DIAMNet | **118.433** | **16.152** | **0.930** | 0.879 | 30.72 |
| Zero | | 237.904 | 35.445 | 0.769 | 0.0 | - |
| Avg | | 235.253 | 60.260 | 0.0 | 0.545 | - |
| VF2 | | 0.0 | 0.0 | 1.0 | 1.0 | $\sim 5 \times 10^3$ |

has been seen previously but the memory of TXL is the representation of the previous segment. In our experiments, the segment size is 64 so that TXL can not learn the global information at a time. A part of the structure information misleads TXL, which is consistent with CNN. A longer segment set for TXL may lead to better results, but it will require much more GPU memory and much longer time for training. RGCN-SUM is much better than RGCN and other sequence models, which shows that sum aggregator is good at modeling vertex representation in this task. The mean aggregator can model the distribution of neighbor but the distribution can also misguide models.

**Effectiveness of the memory.** Table 2 shows the effectiveness of our dynamic attention memory network as the prediction layer. It outperforms the other three pooling methods as well as the simple attention mechanism for all representation architectures. Sum, Mean, and Attention pooling are all

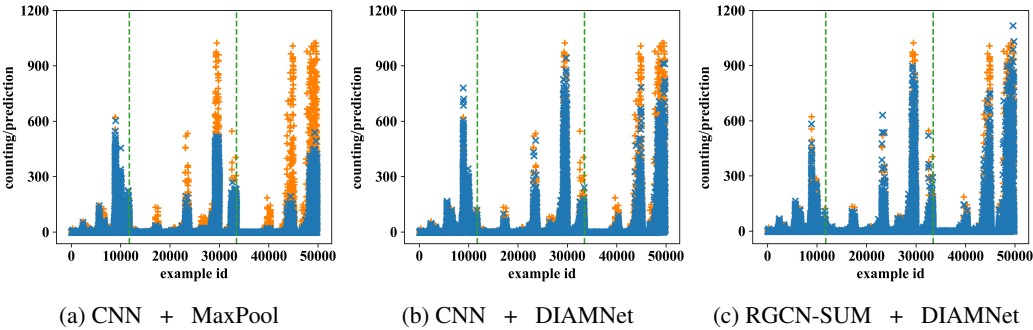

|              |               |                      |
| :----------: | :-----------: | :------------------: |
| (a) CNN + MaxPool | (b) CNN + DIAMNet | (c) RGCN-SUM + DIAMNet |

Figure 5: Model behaviors of three models in *small* dataset. The x-axis is the example id and the y-axis is the count value. We mark the ground truth value as orange + and the predictions as blue ×. We use two green dashed lines to separate patterns into three blocks based on numbers of vertices (3,4, and 8). In each block, the examples are sorted based on the data graphs' sizes.

comparable with each other, because they all gather the global information of the pattern and graph representations. Prediction layer based on max pooling, however, performs the worst, and even worse when the representation layer is CNN or Transformer-XL. This observation indicates that every context of the pattern representation should be counted and we need a better way to compute the weights between each context. The dynamic attention memory with global information of both the pattern and the graph achieves the best results in most of the cases. One of the most interesting observations is that it can even help extract the context of pattern and graph while the representation layer (such as CNN) does not perform very well, which proves the power of our proposed method of DIAMNet.

**Performance on larger graphs.** Table 3 shows our models can be applied to larger-scale graphs. For the *large* dataset, we only choose the best pooling method for each of the baselines to report. We can find most of the results are consistent to the *small* dataset, which means RGCN is the best representation method in our task and the dynamic memory is effective. In terms of the running time, all learning based models are much faster than the traditional VF2 algorithm for subgraph isomorphism counting.

**Model behaviors.** As shown in Figure 5, we compare the model behaviors of the best model (RGCN+SUM) and the worst model (CNN), as well as the great improvement of CNN when memory is added. We can find that CNN+SumPool tends to predict the count value below 400 and has the same behavior between three patterns. This results may come from the fact that CNN can only extract local information of a sequence and Sum pooling is not a good way to aggregate it. However, the memory can memorize local information to each memory cell so it can improve the representation power of CNN and can gain a better performance. RGCN, on the other hand, can better represent the graph structure, so it achieves a better result, especially on the largest pattern (the third block of each figure) compared with CNN. More results can be found in Appendix F.

## 6 CONCLUSIONS

In this paper, we study the challenging subgraph isomorphism counting problem. With the help of deep graph representation learning, we are able to convert the NP-complete problem to a learning based problem. Then we can use the learned model to predict the subgraph isomorphism counts in polynomial time. Counting problem is more related to a global inference rather than only learning node or edge representations. Therefore, we have developed a dynamic intermedium attention memory network to memorize local information and summarize for the global output. We build two datasets to evaluate different representation learning models and global inference models. Results show that learning based method is a promising direction for subgraph isomorphism detection and counting and memory networks indeed help the global inference. We also performed detailed analysis of model behaviors for different pattern and graph sizes and labels. Results show that there is much space to improve when the vertex label size is large. Moreover, we have seen the potential real-world applications of subgraph isomorphism counting problems such as question answering and information retrieval. It would be very interesting to see the domain adaptation power of our developed pretrained models on more real-world applications.

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

## A    LEXICOGRAPHIC ORDER OF CODES

The lexicographic order is a linear order defined as follows:

If $A = (a_0, a_1, \cdots, a_m)$ and $B = (b_0, b_1, \cdots, b_n)$ are the codes, then $A \leq B$ iff either of the following is true:

1. $\exists t, 0 \leq t \leq min(m, n), \forall k < t, a_k = b_k, a_t \prec_e b_t,$
2. $\forall 0 \leq k \leq m, a_k = b_k,$ and $n \geq m.$

In our setting, $a_i = (u_{a_i}, v_{a_i}, \mathcal{X}(u_{a_i}), y_{a_i}, \mathcal{X}(v_{a_i})) \prec_e b_j = (u_{b_j}, v_{b_j}, \mathcal{X}(u_{b_j}), y_{b_j}, \mathcal{X}(v_{b_j}))$ iff one of the following is true:

1. $u_{a_i} < u_{b_j},$
2. $u_{a_i} = u_{b_j}, v_{a_i} < v_{b_j},$
3. $u_{a_i} = u_{b_j}, v_{a_i} = v_{b_j}, y_{a_i} < y_{b_j}.$

## B    PATTERN GENERATOR AND GRAPH GENERATOR

As proposed in Section 5.1, two generators are required to generate datasets. The algorithm about the pattern generator is shown in Algorithm 1. The algorithm first uniformly generates a directed tree. Then it adds the remaining edges with random labels. Vertex labels and edge labels are also uniformly generated but each label is required to appear at least once. Algorithm 2 shows the process of graph generation. Two hyperparameters control the density of subisomorphisms: (1) $\alpha \in [0, 1]$ decides the probability of adding subisomorphisms rather than random edges; (2) $\beta \in \mathbb{N}^+$ is the parameter of Dirichlet distribution to sample sizes of components. After generating several directed trees and satisfying the vertex number requirement, the algorithm starts to add remaining edges. It can add edges in one component and try to add subgraph isomorphisms, or it can randomly add edges between two components or in one component. The following merge subroutine aims to merge these components into a large graph. Shuffling is also required to make datasets hard to be hacked. The search of subisomorphisms in the whole graph is equivalent to the search in components respectively because edges between any two components do not satisfy the necessary conditions.

---

**Algorithm 1** Pattern Generator.

---

**Input:** the number of vertices $N_v$, the number of edges $N_e$, the number of vertex labels $L_v$, the number of edge labels $L_e$.
1: $\mathcal{P} :=$ GenerateDirectedTree($N_v$)
2: AssignNodesLabels($\mathcal{P}$, $L_v$)
3: AddRandomEdges($\mathcal{P}$, $\mathcal{P}$, $null$, $N_e - N_v + 1$)
4: AssignEdgesLabels($\mathcal{P}$, $L_e$)
**Output:** the generated pattern $\mathcal{P}$

---

In Algorithm 1 and Algorithm 2, the function **AddRandomEdges** adds required edges from one component to the other without generating new subgraph isomorphisms. The two component can also be the same one, which means to add in one component. The NEC tree is utilized in Tur-boISO (Han et al., 2013) to explore the candidate region for further matching. It takes $\mathcal{O}(|\mathcal{V}_p|^2)$ time but can significant reduce the searching space in the data graph. It records the equivalence classes and necessary conditions of the pattern. We make sure edges between two components dissatisfy necessary conditions in the NEC tree when adding random edges between them. This data structure and this idea help us to generate more data and search subisomorphisms compared with random generation and traditional subgraph isomorphism searching.

## C    DETAILS OF DATASETS

We can generate as many examples as possible using two graph generators. However, we limit the numbers of training, dev, and test examples whether learning based models can generalize to

---

**Algorithm 2** Graph Generator.

---

**Input:** a pattern $\mathcal{P}$, the number of vertices $N_v$, the number of edges $N_e$, the number of vertex labels $L_v$, the number of edge labels $L_e$, hyperparameters $\alpha$ and $\beta$.

1: $Ns :=$ DirichletSampling($N_v$, $\beta$)
2: $\mathcal{G}s := \{\}$
3: $d_e := N_e$
4: **for** each $n$ in $Ns$ **do**
5:     $g :=$ GenerateDirectedTree($n$)
6:     AssignNodesLabels($g$, $n$)
7:     AssignEdgesLabels($g$, $n - 1$)
8:     $\mathcal{G}s := \mathcal{G}s + \{g\}$
9:     $d_e := d_e - n + 1$
10: **end for**
11: $T_p :=$ RewriteNECTree($\mathcal{P}$)
12: $N_{e,\mathcal{P}} :=$ EdgeCount($\mathcal{P}$)
13: **while** $d_e > 0$ **do**
14:     $g_1, g_2 =$ RandomPick($\mathcal{G}s$, 2)                              ▷ Pick two components randomly
15:     $r =$ RandomNum(0, 1)
16:     **if** $d_e < N_{e,\mathcal{P}}$ **then**
17:         $d_e := d_e$ - AddRandomEdges($g_1$, $g_2$, $T_p$, $d_e$)        ▷ Add $d_e$ edges between $g_1$ and $g_2$
18:     **else**
19:         **if** $r < \alpha$ **then**
20:             $d_e := d_e$ - AddPatterns($g_1$, $\mathcal{P}$)          ▷ Add necessary edges in $g_1$ to add patterns
21:         **else**
22:             $d_e := d_e$ - AddRandomEdges($g_1$, $g_2$, $T_p$, $N_{e,\mathcal{P}}$)
23:         **end if**
24:     **end if**
25: **end while**
26: $\mathcal{G}, f :=$ MergeComponents($\mathcal{G}s$)                          ▷ $f$ is the graph id mapping
27: $\mathcal{G}, f' :=$ ShuffleGraph($\mathcal{G}$)                              ▷ $f'$ is the shuffled graph id mapping
28: $\mathcal{I}s := \{\}$
29: **for** each $g$ in $\mathcal{G}s$ **do**
30:     $Is :=$ SearchSubIsomorphisms($g$, $\mathcal{P}$)
31:     **for** each $I$ in $Is$ **do**
32:         $I :=$ UpdateID($I$, $f$, $f'$)
33:         $\mathcal{I}s := \mathcal{I}s + \{I\}$
34:     **end for**
35: **end for**
**Output:** the generated graph $\mathcal{G}$, the subgraph isomorphisms $\mathcal{I}s$

---

new unseen examples. Pattern and data graphs in different scales have been generated. Parameters for two generators and constraints are listed in Table 4. Two constraints are added to make this task easier. The average degree constraint ($\frac{N_e}{N_v} \leq 4$) is used to keep graphs not so dense, and the subisomorphism counting constraints ($\{c \in \mathbb{N} | c \leq 1024\}$ for *small*, $\{c \in \mathbb{N} | c \leq 4096\}$ for *large*) ensure the difficulty of two datasets are different.

We use a server with an 8-core Intel E5-2620v4 CPU (16 threads) and 512GB RAM to generate and record running time in parallel. The distributions of countings of two datasets are shown in Figure 6. From the figure we can see that both datasets follow the long tail distributions.

# D MORE IMPLEMENTATION DETAILS

In this section, we provide more implementation details other than the representation architectures and the DIAMNet modules.

Table 4: Parameters and constraints for two datasets

|         | Hyperparameters | *small* | *large* |
|---------|-----------------|---------|---------|
| Pattern | $N_v$ | {3, 4, 8} | {3, 4, 8, 16} |
|         | $N_e$ | {2, 4, 8} | {2, 4, 8, 16} |
|         | $L_v$ | {2, 4, 8} | {2, 4, 8, 16} |
|         | $L_e$ | {2, 4, 8} | {2, 4, 8, 16} |
| Graph   | $N_v$ | {8, 16, 32, 64} | {64, 128, 256, 512} |
|         | $N_e$ | {8, 16, $\cdots$, 256} | {64, 128, $\cdots$, 2,048} |
|         | $L_v$ | {4, 8, 16} | {16, 32, 64} |
|         | $L_e$ | {4, 8, 16} | {16, 32, 64} |
|         | $\alpha$ | {0.1, 0.2, $\cdots$, 0.8} | {0.05, 0.1, 0.15} |
|         | $\beta$ | {512} | {512} |

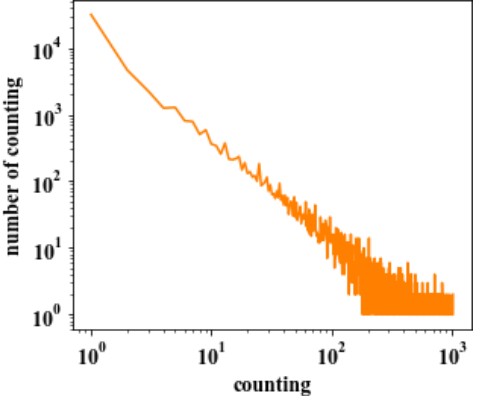

(a) Test data distribution of the *small* dataset.

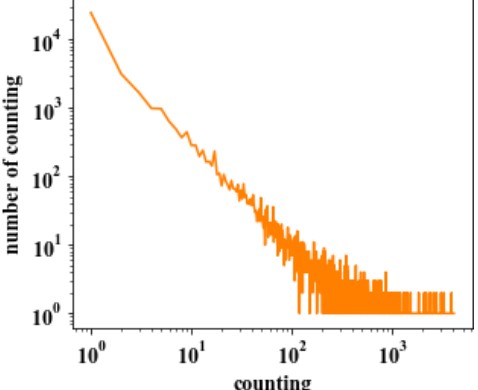

(b) Test data distribution of the *large* dataset.

Figure 6: Subgraph isomorphism counting distributions of two datasets.

### D.1 FILTER NETWORK

Intuitively, not all vertices and edges in graphs can match certain subisomorphisms so we simply add a **FilterNet** to adjust graph encoding as follows:

$$\overline{\boldsymbol{p}} = MaxPool(\boldsymbol{P}), \tag{7}$$

$$\widehat{\boldsymbol{G}} = \boldsymbol{G}\boldsymbol{W}_G^\top, \tag{8}$$

$$\boldsymbol{f}_i = \sigma(\boldsymbol{W}_F(\widehat{\boldsymbol{G}}_{i,:} \odot \overline{\boldsymbol{p}})), \tag{9}$$

$$\boldsymbol{G}_{i,:} = \boldsymbol{f}_i \odot \boldsymbol{G}_{i,:}, \tag{10}$$

where $\widehat{\boldsymbol{G}}$ is the graph representation in the pattern space, $\sigma$ is the sigmoid function, $f_i$ is the gate that decides to filter the $j^{th}$ vertex or the $j^{th}$ edge, $\boldsymbol{W}_G \in \mathbb{R}^{d_p \times d_g}$ and $\boldsymbol{W}_F \in \mathbb{R}^{1 \times d_p}$ are trainable. Thanks to the multi-hot encoding, We can simply use Eq. (7) to accumulate label information of patterns. After this filter layer, only relevant parts of the graphs will be passed to the next representation layer.

### D.2 ATTNPOOL

The computation process of the AttnPool (Mean) is:

$$\check{\boldsymbol{p}} = MeanPool(\check{\boldsymbol{P}}), \tag{11}$$

$$\check{\boldsymbol{g}} = MultiHead(\widetilde{\boldsymbol{p}}, \check{\boldsymbol{G}}, \check{\boldsymbol{G}}). \tag{12}$$

Combing with the pooling strategy to make $\check{P}$ as $\widetilde{P} \in \mathbb{R}^{1 \times d}$, the time complexity is decreased to $\mathcal{O}(|\mathcal{V}_P| + |\mathcal{V}_G|)$ or $\mathcal{O}(|\mathcal{E}_P| + |\mathcal{E}_G|)$. We also implement three variants with sum-pooling, mean-

pooling, and max-pooling respectively. Results of AttnPool (Sum) and AttnPool (Max) are shown in the Appendix E.

### D.3 DIAMNET INITIALIZATION

There are many ways to initialize the memory $\{\boldsymbol{m}_1^{(0)}, \cdots, \boldsymbol{m}_M^{(0)}\}$. In experiments, we simply initialize $\boldsymbol{m}_i^{(0)}$ by

$$s = \lfloor \frac{|\mathcal{V}_G|}{M} \rfloor, \tag{13}$$

$$k = |\mathcal{V}_G| - (M - 1) \cdot s, \tag{14}$$

$$\boldsymbol{m}_i^{(0)} = MeanPool(\{\check{G}_{i \cdot s}, \cdots, \check{G}_{i \cdot s + k - 1}\}), \tag{15}$$

where $s$ is the stride and $k$ is the kernel size. In Table 5, we compared two additional pooling methods with MeanPool in Eq. (15).

Table 5: Detailed results of different models with different predict networks on the *small* dataset.

| | Models | Test | | | | | | | |
|---|---|---|---|---|---|---|---|---|---|
| | | RMSE | MAE | $P_{zero}$ | $R_{zero}$ | $F1_{zero}$ | $P_{nonzero}$ | $R_{nonzero}$ | $F1_{nonzero}$ |
| CNN | SumPool | 55.429 | 11.057 | 0.997 | 0.678 | 0.807 | 0.249 | 0.980 | 0.397 |
| | MeanPool | 57.298 | 10.517 | **0.999** | 0.697 | 0.821 | 0.312 | **0.995** | 0.475 |
| | MaxPool | 47.353 | 11.212 | 0.993 | 0.715 | 0.832 | 0.372 | 0.973 | 0.539 |
| | AttnPool (Sum) | 57.800 | 15.642 | 0.753 | 0.825 | 0.787 | 0.747 | 0.655 | 0.698 |
| | AttnPool (Mean) | 55.963 | 12.717 | 0.991 | 0.665 | 0.796 | 0.207 | 0.937 | 0.340 |
| | AttnPool (Max) | 57.102 | 11.196 | 0.989 | 0.682 | 0.807 | 0.267 | 0.936 | 0.416 |
| | DIAMNet (SumInit) | 43.312 | 10.284 | 0.900 | **0.871** | **0.885** | **0.789** | 0.833 | **0.810** |
| | DIAMNet (MeanInit) | **34.448** | **6.953** | 0.980 | 0.805 | 0.884 | 0.623 | 0.952 | 0.753 |
| | DIAMNet (MaxInit) | 35.055 | 7.188 | 0.980 | 0.802 | 0.882 | 0.617 | 0.951 | 0.748 |
| RNN | SumPool | 29.955 | 5.740 | 0.979 | 0.846 | 0.908 | 0.717 | 0.956 | 0.819 |
| | MeanPool | 31.010 | 6.447 | 0.971 | 0.859 | 0.912 | 0.746 | 0.942 | 0.833 |
| | MaxPool | 30.824 | 6.236 | **0.994** | 0.771 | 0.869 | 0.532 | **0.982** | 0.690 |
| | AttnPool (Sum) | 32.481 | 6.262 | 0.977 | 0.853 | 0.911 | 0.733 | 0.953 | 0.829 |
| | AttnPool (Mean) | 31.857 | 6.025 | 0.985 | 0.827 | 0.899 | 0.672 | 0.967 | 0.793 |
| | AttnPool (Max) | 41.186 | 11.011 | 0.948 | 0.888 | 0.917 | 0.811 | 0.908 | 0.857 |
| | DIAMNet (SumInit) | **29.682** | 5.632 | 0.925 | **0.949** | 0.937 | **0.921** | 0.885 | **0.903** |
| | DIAMNet (MeanInit) | 29.743 | **5.547** | 0.948 | 0.907 | 0.927 | 0.846 | 0.911 | 0.877 |
| | DIAMNet (MaxInit) | 29.778 | 5.827 | 0.951 | 0.926 | **0.938** | 0.879 | 0.919 | 0.899 |
| TXL | SumPool | 34.391 | 7.042 | **0.976** | 0.833 | 0.899 | 0.690 | **0.947** | 0.798 |
| | MeanPool | 32.569 | **6.656** | 0.972 | 0.808 | 0.882 | 0.632 | 0.935 | 0.754 |
| | MaxPool | 65.152 | 30.289 | 0.406 | **0.965** | 0.572 | **0.977** | 0.509 | 0.669 |
| | AttnPool (Sum) | 55.166 | 12.748 | 0.870 | 0.832 | 0.851 | 0.721 | 0.778 | 0.748 |
| | AttnPool (Mean) | 37.721 | 7.426 | 0.965 | 0.790 | 0.869 | 0.592 | 0.914 | 0.719 |
| | AttnPool (Max) | 60.196 | 18.916 | 0.529 | 0.892 | 0.664 | 0.898 | 0.545 | 0.679 |
| | DIAMNet (SumInit) | 32.659 | 6.904 | 0.907 | 0.940 | **0.923** | 0.908 | 0.861 | **0.884** |
| | DIAMNet (MeanInit) | **31.649** | 6.680 | 0.961 | 0.847 | 0.900 | 0.725 | 0.921 | 0.811 |
| | DIAMNet (MaxInit) | 37.877 | 8.317 | 0.914 | 0.884 | 0.899 | 0.810 | 0.856 | 0.832 |
| RGCN | SumPool | 32.414 | 6.578 | 0.984 | 0.829 | 0.900 | 0.677 | 0.965 | 0.796 |
| | MeanPool | 33.829 | 7.152 | 0.981 | 0.795 | 0.878 | 0.599 | 0.951 | 0.735 |
| | MaxPool | 50.851 | 9.707 | 0.982 | 0.780 | 0.869 | 0.559 | 0.952 | 0.704 |
| | AttnPool (Sum) | 33.816 | 6.834 | 0.991 | 0.809 | 0.890 | 0.628 | 0.977 | 0.764 |
| | AttnPool (Mean) | 32.526 | 6.523 | **0.994** | 0.774 | 0.870 | 0.540 | **0.982** | 0.697 |
| | AttnPool (Max) | 42.970 | 8.850 | 0.964 | 0.847 | 0.902 | 0.724 | 0.928 | 0.813 |
| | DIAMNet (SumInit) | 29.459 | 5.688 | 0.964 | 0.871 | 0.915 | 0.772 | 0.931 | 0.844 |
| | DIAMNet (MeanInit) | **28.712** | 5.782 | 0.924 | 0.912 | 0.918 | 0.859 | 0.877 | 0.868 |
| | DIAMNet (MaxInit) | 29.345 | **5.501** | 0.937 | **0.918** | **0.927** | **0.866** | 0.896 | **0.881** |
| RGCN-SUM | SumPool | 22.379 | 3.958 | 0.988 | 0.860 | 0.920 | 0.746 | 0.974 | 0.845 |
| | MeanPool | 22.483 | 4.254 | 0.987 | 0.833 | 0.903 | 0.685 | 0.971 | 0.803 |
| | MaxPool | 42.434 | 7.900 | 0.994 | 0.780 | 0.874 | 0.554 | 0.984 | 0.709 |
| | AttnPool (Sum) | 23.214 | 4.149 | 0.992 | 0.830 | 0.904 | 0.679 | 0.981 | 0.802 |
| | AttnPool (Mean) | 24.875 | 5.131 | **0.998** | 0.725 | 0.840 | 0.399 | **0.991** | 0.569 |
| | AttnPool (Max) | 35.037 | 6.845 | 0.987 | 0.794 | 0.880 | 0.593 | 0.967 | 0.735 |
| | DIAMNet (SumInit) | **21.385** | **3.811** | 0.962 | 0.886 | 0.922 | 0.803 | 0.930 | 0.862 |
| | DIAMNet (MeanInit) | 21.734 | 3.853 | 0.964 | **0.887** | **0.924** | **0.804** | 0.934 | **0.864** |
| | DIAMNet (MaxInit) | 23.940 | 4.156 | 0.966 | 0.882 | 0.922 | 0.794 | 0.937 | 0.860 |
| Zero | | 67.195 | 13.716 | 1.0 | 0.614 | 0.761 | 0.0 | 0.0 | 0.0 |
| Avg | | 65.780 | 21.986 | 0.0 | 0.0 | 0.0 | 1.0 | 0.386 | 0.557 |
| VF2 | | 0.0 | 0.0 | 1.0 | 1.0 | 1.0 | 1.0 | 1.0 | 1.0 |

## E    DETAILED RESULTS OF DIFFERENT PREDICTNETS

Simple pooling, attention based pooing, and attention with memory can be used to fuse pattern and graph representations in our framework. AttnPool (Sum) and AttnPool (Max) can be regarded as two variants of AttnPool (Mean) by replacing pooling methods. Table 5 shows results of different representation models with different interaction networks in the *small* dataset. AttnPool (Mean) and DIAMNet (MeanInit) usually perform better compared with other pooling methods.

## F    FURTHER DISCUSSIONS

As shown in Figure 7, different interaction modules perform differently in different views. We can find MaxPool always predicts higher counting values when the pattern is small and the graph is large, while AttnPool always predicts very small numbers except when the pattern vertex size is 8, and the graph vertex size is 64. The same result appears when we use edge sizes as the x-axis. This observation shows that AttnPool has difficulties predicting counting values when either of the pattern and the graph is small. It shows that attention focuses more on the zero vector we added rather than the pattern pooling result. Our DIAMNet, however, performs the best in all pattern/graph sizes. When the bins are ordered by vertex label sizes or edge label sizes, the performance of all the three interaction modules among the distribution are similar. When bins are ordered by vertex label sizes, we have the same discovery that AttnPool prefers to predict zeros when then patterns are small. MaxPool fails when facing complex patterns with more vertex labels. DIAMNet also performs not so good over these patterns. As for edge labels, results look good for MaxPool and DIAMNet but AttnPool is not satisfactory.

As shown in Figure 8, different representation modules perform differently in different views. CNN performs badly when the graph size is large (shown in Figure 8a and 8d) and patterns become complicated (show in Figure 8g and 8j), which further indicates that CNN can only extract the local information and suffers from issues when global information is need in larger graphs. RNN, on the other hand, performs worse when the graph are large, especially when patterns are small (show in Figure 8e), which is consistent with its nature, intuitively. On the contrary, RGCN-SUM with DIAMNet is not affected by the edge sizes because it directly learns vertex representations rather than edge representations.

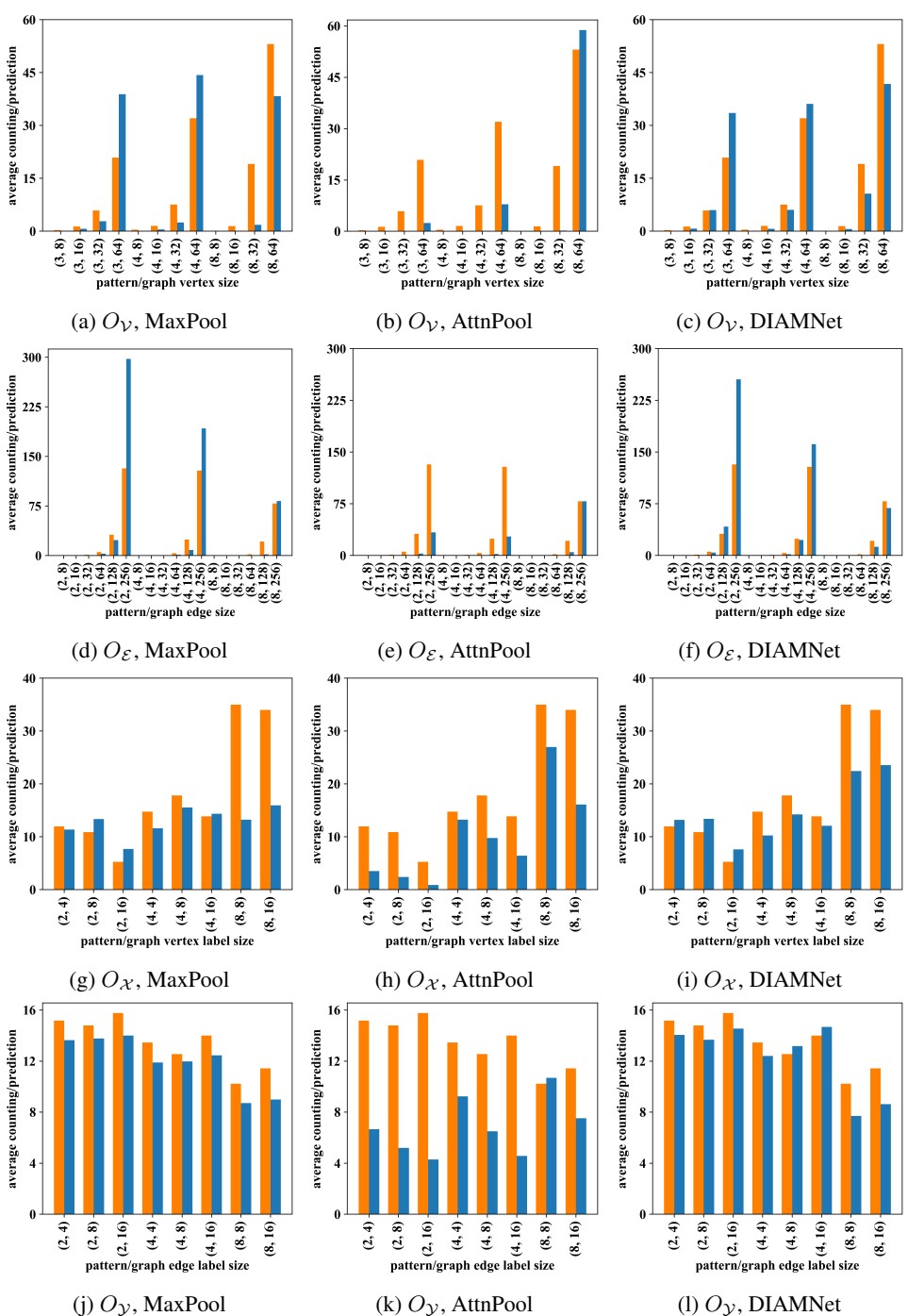

Figure 7: Model behaviors of three CNN based models on the *small* dataset. We compare MaxPool, AttnPool, and DIAMNet in four different view settings. Here $O_\mathcal{V}$ means that on the x-axis the bins are ordered by the size of pattern/graph vertex, $O_\mathcal{E}$ by the size of edge, $O_\mathcal{X}$ by the size of vertex label, and $O_\mathcal{Y}$ by the size of edge label. The orange color refers to the ground truth and the blue color refers to the predictions.

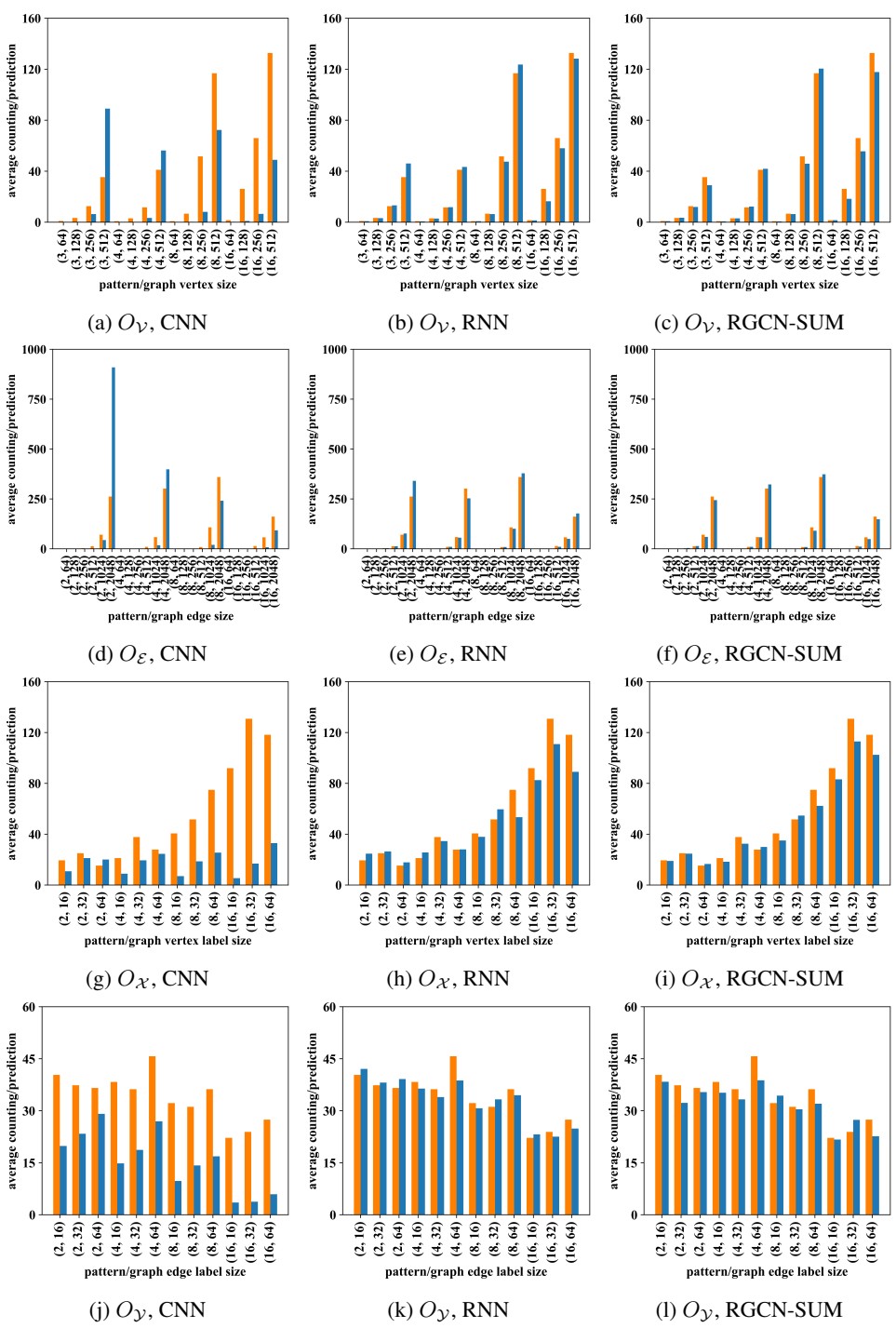

Figure 8: Model behaviors of three models on the *large* dataset. We compare CNN, RNN, and RGCN-SUM with our DIAMNet in four different view settings. Here $O_\mathcal{V}$ means that on the x-axis the bins are ordered by the size of pattern/graph vertex, $O_\mathcal{E}$ by the size of edge, $O_\mathcal{X}$ by the size of vertex label, and $O_\mathcal{Y}$ by the size of edge label. The orange color refers to the ground truth and the blue color refers to the predictions.

