# OpenReview forum: "Neural Subgraph Isomorphism Counting"
_ICLR.cc/2020/Conference — Reject_

### Official Review · AnonReviewer3 · 2019-10-21
**Official Blind Review #3**

**Rating:** 3

**Review:**

The paper proposed NN based subgraph counting. By using synthetically generated graphs, NN learns the number of occurences of a given queried graph called 'pattern'. The author proposes a specific architecture for learning the count based on the multi-head attention method. The authors empirically evaluated the performance on the synthetic dataset.

The problem setting would be interesting. Applying NN to counting a subgraph is novel as far as I know. My current concerns are mainly on the appropriateness of the experimental evaluation.

In the tables, the trivial baseline 'Zero' is shown as F1_zero = 0, but is this correct? I think this should be non-zero. If zero is 'positive' in F1_zero, recall is 1 and precision is 0.75 (because the author set 75% of data as zero). F-score is harmonic mean of them, which is 0.86.

RMSE and MAE of the Zero prediction is shown, but the more standard baseline of the error would be a constant prediction (e.g., the average of test points is often used, which can evaluate how much variance can be explained by the model).

Why were 75 percent of countings set as 0 in the evaluation dataset? This rate is seemingly a bit large for the evaluation purpose. I guess that when this percentage is much more smaller, MSE would increase. In other words, current MSE/MAE values might be underestiamted compared with when all the test points have non-zero countings.

The evaluation is only for synthetic dataset for which generating process is designed by the authors. If possible, evaluation on benchmark graph datasets would be convincing though creating the ground truth might be difficult for larger graphs.

Minor comment:
At the third line of Sec 3.2: '|V_G| is the number of pattern nodes' should be |V_P|.


**Experience Assessment:**

I do not know much about this area.

**Review Assessment: Checking Correctness Of Derivations And Theory:**

I did not assess the derivations or theory.

**Review Assessment: Checking Correctness Of Experiments:**

I assessed the sensibility of the experiments.

**Review Assessment: Thoroughness In Paper Reading:**

I read the paper at least twice and used my best judgement in assessing the paper.

---

> ### Author Response · Authors · 2019-11-13
> **Response to Review #3**
>
> Thanks for your questions.
>
> 1) Thanks for correcting the Zero baseline. Corrected precision, recall, and F1 scores have been updated.
>
> 2) The constant prediction, e.g., the average count of training data, is also added in the latest version as well as follows.
>
> Small
>         |  RMSE  | MAE     | F1_0   | F1_nonzero
> Zero | 67.195 | 13.716 | 0.761 | 0.0
> Avg  | 65.780 | 21.986  | 0.0     | 0.557
>
> Large
>          |  RMSE    | MAE    | F1_0   | F1_nonzero
> Zero | 237.904 | 35.445 | 0.769 | 0.0
> Avg  | 235.253 | 60.260 | 0.0      | 0.545
>
> The average of training data is very close to that of test data. But using the average of training data is fairer than using the average of test data. This baseline (Avg) is worse than Zero.
>
> 3) “75% zero counting graphs”. This setting is designed in purpose to evaluate neural models. As traditional algorithms do not have problems when there is no subgraph isomorphism detected in a data graph, neural models would fit the training data. However, in practice, there will be a lot of applications that zero counting exists for most of the cases. Therefore, we also add a lot of zero counting data. In addition, we can also get some sense of the performance when evaluating the F1_0 and F1_nonzero to compare different models, although we are not building a binary classifier. It may be possible to set different percentages of zero counting data. However, the results will be similar in terms of relative performance. We also agree that current MSE/MAE values might be underestimated compared with when all the test points have non-zero countings, and this also means that we are challenging ourselves and the community with a more difficult problem.
>
> 4) Because this problem is NP-complete, there is no suitable dataset for both traditional algorithms and neural algorithms. Traditional algorithms are hard to scale to large graphs while neural models require plenty of data. But we agree with you that benchmark datasets are more convincing. We will release our generation code and learning code as one benchmark for future research.

---

### Official Review · AnonReviewer2 · 2019-10-22
**Official Blind Review #2**

**Rating:** 3

**Review:**

This paper proposes a method called Dynamic Intermedium Attention Memory Network (DIAMNet) to learn the subgraph isomorphism counting for a given pattern graph P and target graph G. This requires global information unlike usual GNN cases such as node classification, link prediction, community detection. First, input graphs P and G are converted embedding vectors through sequence models (CNN, RNN, Transformer-XL) or graph models (RGCN), and fed into their DIAMNet that uses an external memory as an intermedium to attend both the pattern and the graph. The external memory is updated based on multi-head attention as in Transformer. The output of DIAMNet is passed to FC that outputs 'count' directly. The training is based on minimizing MSE loss as a regression problem. Extensive experimental evaluations report that DIAMNet showed superior performance over competing methods and baselines.

This paper targets subgraph isomorphism counting as a learning problem for the first time I guess, and the proposed method combined with both graph- and sequence-based encoding is technically interesting. However, there are still two major issues of 1) why counting? 2) the RMSE loss for regression on counts 3) baseline of 'Zero'.

1) the most unclear point is 'why counting?'. If I understand it, this method can be applied to subgraph isomorphism (NP-hard) or graph isomorphism (unknown complexity) as binary classification, and experimental evaluations can use the datasets used in evaluating VF2 or Naughty. It would be better to start this fundamental problem that would have many clear applications. Compared to subgraph isomorphism or graph isomorphism, the need for knowing accurate 'counts' of subgraph isomorphisms is unconvincing (given that we cannot explicitly obtains all subgraph matchings). Note that there is some existing research on GNNs targeted 'graph matching' and 'graph similarity'.

Also, the used datasets intentionally restrict the possible values for the number of subgraph isomorphisms, but the counts would be exponentially large if we consider practical (dense) graphs.

2) the method fits the model using (R)MSE loss, but minimizing log errors ((R)MSLE) would be better considering distributions of response values (counts) of the used datasets in Figure 6. Fitting the MSE loss is not good for such highly skewed cases, and for example, might focus only on the few instances having very large count values. Or, if such instances are very small, training ignores all such extreme instances. Either way would be questionable when we consider learning 'subgraph isomorphism counting' in general.

Also, the error of counts by MSE or MAE would be less informative and it would be unclear how much errors are tolerant in practical use cases of this method.

3) To interpret the RMSE and MAE values, Table 2 has the value for 'Zero'. This is for a constant predictor always returning zeros for any inputs. However, given that the loss is MSE, constant prediction values should be the average counts in the training data, not zero.


**Experience Assessment:**

I have published one or two papers in this area.

**Review Assessment: Checking Correctness Of Derivations And Theory:**

N/A

**Review Assessment: Checking Correctness Of Experiments:**

I carefully checked the experiments.

**Review Assessment: Thoroughness In Paper Reading:**

I read the paper thoroughly.

---

> ### Author Response · Authors · 2019-11-13
> **Response to Review #2**
>
> Thanks for your questions.
>
> Q.1
> Q.1.1. Besides what we explained in the “general response of why counting”, we would like to emphasize that simply using a binary classifier for subgraph isomorphism and graph isomorphism would be less useful than counting in knowledge discovery and "how many" based KBQA, although the same representation and ways of representation learning could be applied.
>
> Q.1.2. “Note that there is some existing research on GNNs targeted 'graph matching' and 'graph similarity'.”
> Yes, we have cited some of these existing works in the related work section. However, for subgraph isomorphism counting, we need different types of graph encoding, which is shown in sections 4.1.1 and 4.2.1.
>
> Q.1.3. “the used datasets intentionally restrict the possible values for the number of subgraph isomorphisms, but the counts would be exponentially large if we consider practical (dense) graphs.”
> It is true for homogeneous patterns querying dense graphs, but in practice, heterogeneous patterns with node and edge types are more useful, e.g., to query knowledge graphs with node and edge types.
>
> 2) The objectives based on (R)MSE or (R)MAE correspond to a regression loss. Although the labels seem to be skewed, given the power of deep representation learning, it will be able to map the graph and pattern pair in a semantic space that can better perform regression.  Log errors are not good because final models cannot handle complex cases (whose counts are large). Errors between log predictions and log counts are also not suitable because predictions at the early training steps can be negative. If we simply use ReLU (in prediction) when computing losses, models are easy to get stuck in a local optimum to predict zero all the time. We have tried all the above options but training processes did not even converge.
>
> We constrain counts for the computational time of traditional algorithms and the interpretability of errors. Traditional algorithms will spend much more time on complex graphs. Errors are easy to be disturbed by those cases. In our datasets, we limit the count <= 1024 when |E| <= 256 and the count <= 4096 when |E| <= 2048.
>
> 3) We use Zero as one of our baselines because it is a local optimum. We have added the average count of training data as our baseline in the latest submission as well as follows. The average count is worse than Zero prediction.
>
> Small
>         |  RMSE  | MAE     | F1_0   | F1_nonzero
> Zero | 67.195 | 13.716 | 0.761 | 0.0
> Avg  | 65.780 | 21.986  | 0.0     | 0.557
>
> Large
>          |  RMSE    | MAE    | F1_0   | F1_nonzero
> Zero | 237.904 | 35.445 | 0.769 | 0.0
> Avg  | 235.253 | 60.260 | 0.0      | 0.545

---

### Official Review · AnonReviewer4 · 2019-11-06
**Official Blind Review #4**

**Rating:** 3

**Review:**

This paper studied how to leverage the power of graph neural networks for counting subgraph isomorphism. The motivation is that the current subgraph isomorphism detection is NP-complete problem and a proposed approach based on GNN could approximately solve the counting problem in polynomial time. Then they relaxed original subgraph isomorphism (which is equivalent to the exact subgraph matching problem)  and proposed the problem of doing subgraph isomorphism counting task. The GNN and sequence modeling methods are discussed for solving this problem. The experimental results confirmed the effectiveness of these methods.

Although I found the subgraph isomorphism counting problem is an interesting problem, I did not know how much practical usefulness of this task. More practical use case would be search for the matched subgraphs given the sub-graph query using subgraph isomorphism detection.

Also, although authors mentioned some approximation systems/methods in graph database community such as TurboISO (Han et al., 2013), VF3 (Carletti et al., 2018), and other approximation techniques [1][2], authors did not consider them as baselines to compare. These methods may also have limitations to deal with real-large graph but for the graph size that this paper studied I think they are fine to deal with. A parallel issue is that GNN also has scalability issues as well when dealing with large graphs [3]. Without comparing these existing fast (approximation) methods, it is really unfair to compare with only non-DL baseline VF2, which seems served as ground-truth as well.

[1]  A Neural Graph Isomorphism Algorithm Based on Local Invariants, ESANN'2003
[2] Subgraph Isomorphism in Polynomial Time
[3] FastGCN: Fast Learning with Graph Convolutional Networks via Importance Sampling

In terms of technical contributions, they leverage some existing sequence models (CNN, RNN and so on) and graph models (RGNN) and the whole framework is similar to doing a graph matching networks (without considering node alignment) for a regression task. The DYNAMIC INTERMEDIUM ATTENTION MEMORY NETWORK is interesting yet simple. I am not entirely clear what's the output  of this interactional module. The figure 4 shows the overall architecture of subgraph isomorphism counting model, which needs better descriptions to understand exact input and output for each module. In general, the novelty of this part is incremental.

Finally, this subgraph isomorphism counting problem is closely related to graphlet counting problem. In the paper, the subgraph pattern considered seems like almost identical to graphlets the previous research extensively studied. I did not see any discussion about the connection of these two tasks either.

Minor comments:

|V_G| is is the number of pattern nodes -> |V_p| is is the number of pattern nodes

**Experience Assessment:**

I have published in this field for several years.

**Review Assessment: Checking Correctness Of Derivations And Theory:**

I assessed the sensibility of the derivations and theory.

**Review Assessment: Checking Correctness Of Experiments:**

I carefully checked the experiments.

**Review Assessment: Thoroughness In Paper Reading:**

I read the paper at least twice and used my best judgement in assessing the paper.

---

> ### Author Response · Authors · 2019-11-13
> **Response to Review #4**
>
> Thanks for your questions.
>
> 1) Please refer to the general response of “why counting”.
>
> 2) The reason we didn't compare with TurboISO and VF3 is that our graphs are generated by Algorithm 2, where the idea comes from TurboISO and VF3. When generating a graph, we do not need to run traditional algorithms to get the count but to add pattern isomorphisms. Random edges are added following the rule that breaks necessary conditions (Line 20 in Alg 2). These necessary conditions are used in TurboISO and VF3 to find candidate subregions. If we use TurboISO and VF3 as baseline algorithms, we believe the two methods will terminate in a short time. VF2 is considered one of the most representative algorithms and we use it to demonstrate the time of the magnitude of traditional methods.
>
> As for other approximation methods, [Q1] is designed for graph isomorphism rather than subgraph isomorphism; [Q2] is still not suitable due to the exponential space requirement (19 vertices requires 1.2 Mbyte of disk space, shown Page 23). [Q2] only compares their method with Ullman’s algorithm on graphs with 19 vertices and achieves 16 times speedup. However, VF2 is 1,000 times faster than Ullman’s algorithm when |V| > 200. We do not think [Q2] can be applied to our two datasets. Isomorphism and subgraph isomorphism problem cannot be solved by sampling as [Q3], and that’s the reason why RGCN is worse than RGCN-SUM.
>
> [Q1]  A Neural Graph Isomorphism Algorithm Based on Local Invariants, ESANN'2003
> [Q2] Subgraph Isomorphism in Polynomial Time
> [Q3] FastGCN: Fast Learning with Graph Convolutional Networks via Importance Sampling
>
> 3) The output of DIAMNet is the memory itself, where it has M blocks and each block is a d-dimensional vector shown in Section 4.3.
>
> 4) Graphlets are small connected non-isomorphic induced subgraphs (usually 3-5 nodes) of a large network. We want to use neural models to approximately solve a general pattern counting problem. The pattern can be sparse or dense, homogeneous or heterogeneous. As Table 4 shows, we have many diverse structures of both patterns and graphs. This generalization requires a much more powerful ability of inference.

---

### Official Review · AnonReviewer1 · 2019-11-11
**Official Blind Review #1**

**Rating:** 6

**Review:**

This paper proposes a dynamic inter-medium attention memory network and model the sub-graph isomorphism counting problem as a learning problem with both polynomial training and prediction time complexities.
Since the testing time is reported in this paper, and the time complexity is one of the main contribution of this paper. The hardware and software used to run the algorithm should be reported in the main article.

The author argues that if we use neural networks to learn distributed representations for V_G and V_p or \xi_G and \xi_P without self-attention, the computational cost will acceptable for large graphs, but the missing of self-attention will hurt the performance. It’s encouraged to do corresponding experiments to compare it with the proposed method and better support the algorithm.

One of the main advantages of this paper is that the proposed method can efficiently deal with large graph tasks, so the model behaviors of different models in large dataset similar to Figure 5 is encouraged to be given.


**Experience Assessment:**

I have published in this field for several years.

**Review Assessment: Checking Correctness Of Derivations And Theory:**

I carefully checked the derivations and theory.

**Review Assessment: Checking Correctness Of Experiments:**

I carefully checked the experiments.

**Review Assessment: Thoroughness In Paper Reading:**

I read the paper thoroughly.

---

> ### Author Response · Authors · 2019-11-13
> **Response to Review #1**
>
> Thanks for your suggestions.
>
> 1) The hardware information and software information has been added to the latest version. Training and evaluating were finished on one single NVIDIA GTX 1080 Ti GPU under the PyTorch framework.
>
> 2) When the edge size of a graph increases to 256, it is already hard for neural models to do self-attention for graphs. Transformer-XL [1] is proposed to solve the computational cost problem. Generally, a 6-layer Transformer-XL should be better than a 3-layer GRU, but results in Table 2 and Table 3 show that Transformer is worse instead. Subgraph isomorphism counting requires the whole pattern information and the whole graph information.
>
> We can try to implement a model with self-attention and source attention, but it can be only trained in rather small batch sizes and applied to toy data. We think this model cannot be helpful to solve the subgraph isomorphism counting problem.
>
> [1] Z. Dai, Z. Yang, Y. Yang, J. G. Carbonell, Q. V. Le, and R. Salakhutdinov. Transformer-xl: Attentive language models beyond a fixed-length context. In ACL, pp. 2978–2988, 2019.
>
> 3) We have provided 24 more figures and further discussions in Appendix F. Those figures can provide more information about the behaviors of different models with different data.

---

### Author Response · Authors · 2019-11-13
**General response of “why counting”**

Although solving subgraph matching/enumeration can solve subgraph counting but not the other way round, counting itself is still very useful. Counting the number of isomorphic copies has been proven to be useful for bioinformatics [1], [2], chemoinformatics [3], and online social network analysis [4]. Especially, when counting a new structure that a professional may query (e.g., a gene structure, a protein structure, or a social network structure), a first step may be a rough estimation instead of exact finding. Then a fast algorithm to estimate may save a lot of time for such kind of knowledge discovery. Our counting task is also related to graphlet counting in database and data mining fields. However, our framework can count patterns with much more heterogeneous nodes and edges rather than 3-5 nodes in graphlets.

Counting is also an important task especially for knowledge-based question answering (KBQA). More importantly, nowadays, most modern knowledge graphs are stored in RDF graph databases. The schema of such databases are more complex and counting based on the graph is preferred, for example:
Ex (used in our introduction): “how many languages are there in Africa speaking by people living near the banks of the Nile River?”
After semantic parsing, such questions should be mapped to be a subgraph counting problem, where the subgraphs should follow some types of nodes and relations. Therefore, automatically counting can solve a particular KBQA problem in the future. However, to our best knowledge, we haven’t found any existing large-scale KBQA dataset that is specifically developed for the subgraph counting problem. This is why we developed our training and test datasets, which can serve as a pre-training step for higher-order “how many” KBQA problems.

[1] R. Milo, S. Shen-Orr, S. Itzkovitz, N. Kashtan, D. Chklovskii, and U. Alon, “Network motifs: Simple building blocks of complex networks,” Science, vol. 298, no. 5594, pp. 824–827, 2002.
[2] N. Alon, P. Dao, I. Hajirasouliha, F. Hormozdiari, and S. C. Sahinalp, Biomolecular network motif counting and discovery by color coding, Bioinformatics, vol. 24, no. 13, pp. i241–i249, 2008.
[3] J. Huan, W. Wang, and J. Prins, Efficient mining of frequent subgraphs in the presence of isomorphism, ICDM, 2003, p. 549.
[4] M. Kuramochi and G. Karypis, Frequent subgraph discovery, ICDM, 2001, pp. 313–320.

---

### Decision · Program_Chairs · 2019-12-19

**Decision:**

Reject

**Comment:**

This paper proposes a method called Dynamic Intermedium Attention Memory Network (DIAMNet) to learn the subgraph isomorphism counting for a given pattern graph P and target graph G. However, the reviewers think the experimental comparisons are insufficient. Furthermore,  the evaluation is only for synthetic dataset for which generating process is designed by the authors. If possible, evaluation on benchmark graph datasets would be convincing though creating the ground truth might be difficult for larger graphs.